# Plant-Derived Molecule 4-Methylumbelliferone Suppresses FcεRI-Mediated Mast Cell Activation and Allergic Inflammation

**DOI:** 10.3390/molecules27051577

**Published:** 2022-02-27

**Authors:** Hui-Na Wang, Qiu-An Xiang, Hao-Hui Lin, Jie-Ning Chen, Wen-Jie Guo, Wan-Meng Guo, Xiang-Ning Yue, Zhen-Fu Zhao, Kunmei Ji, Jia-Jie Chen

**Affiliations:** Department of Biochemistry and Molecular Biology, School of Basic Medical Sciences, Health Science Center, Shenzhen University, Shenzhen 518060, China; 15238608260@163.com (H.-N.W.); szuxiangqiuan@163.com (Q.-A.X.); linhaohui0414@163.com (H.-H.L.); Jenny_ning134340@163.com (J.-N.C.); guowenjie@163.com (W.-J.G.); miraclegwm@163.com (W.-M.G.); T30420130613@163.com (X.-N.Y.); zhb@szu.edu.cn (Z.-F.Z.)

**Keywords:** 4-methylumbelliferone, mast cell, FcεRI signaling, SYK

## Abstract

Mast cells (MCs) are an important treatment target for high-affinity IgE Fc receptor (FcεRI)-mediated allergic diseases. The plant-derived molecule 4-methylumbelliferone (4-MU) has beneficial effects in animal models of inflammation and autoimmunity diseases. The aim of this study was to examine 4-MU effects on MC activation and probe the underlying molecular mechanism(s). We sensitized rat basophilic leukemia cells (RBLs) and mouse bone marrow-derived mast cells (BMMCs) with anti-dinitrophenol (DNP) immunoglobulin (Ig)E antibodies, stimulated them with exposure to DNP-human serum albumin (HSA), and then treated stimulated cells with 4-MU. Signaling-protein expression was determined by immunoblotting. In vivo allergic responses were examined in IgE-mediated passive cutaneous anaphylaxis (PCA) and ovalbumin (OVA)-induced active systemic anaphylaxis (ASA) mouse models. 4-MU inhibited β-hexosaminidase activity and histamine release dose-dependently in FcεRI-activated RBLs and BMMCs. Additionally, 4-MU reduced cytomorphological elongation and F-actin reorganization while down-regulating IgE/Ag-induced phosphorylation of SYK, NF-κB p65, ERK1/2, p38, and JNK. Moreover, 4-MU attenuated the PCA allergic reaction (i.e., less ear thickening and dye extravasation). Similarly, we found that 4-MU decreased body temperature, serum histamine, and IL4 secretion in OVA-challenged ASA model mice. In conclusion, 4-MU had a suppressing effect on MC activation both in vitro and in vivo and thus may represent a new strategy for treating IgE-mediated allergic conditions.

## 1. Introduction

Mast cells (MCs) are key effector cells in IgE-mediated allergic and inflammatory reactions, including asthma, allergic rhinitis, and atopic dermatitis [1,2]. Allergen-coupled IgE antibodies bind high-affinity IgE Fc receptors (FcεRIs) on the surface of circulating MCs. The resultant IgE/FcεRI pathway activation promotes MC degranulation and the release of bioactive mediators and inflammatory factors, such histamine, interleukin (IL)-6, and IL-13, which mediate allergic reactions [3,4,5]. Therefore, FcεRI-mediated allergic diseases could potentially be managed by pharmacological inhibition of MC activation or degranulation.

The coumarin-derivative 4-methylumbelliferone (4-MU; CAS number 90-33-5), found mainly in *Umbelliferae* and *Asteraceae* plant species, has anti-inflammatory effects [6]. 4-MU is an umbelliferone (molecular formula, C_10_H_8_O_3_) with a methylated carbon at C4 (its chemical structure shown in Figure 1a). Notably, 4-MU is an orally available dietetic product and has been approved as a drug called hymecromone for the treatment of biliary spasms in Europe and Asia [7]. Pharmacologically, 4-MU inhibits the synthesis of hyaluronic acid (HA), a glycosaminoglycan expressed ubiquitously in connective tissues, and it has been implicated in inflammation, autoimmunity, tumor growth, and metastasis [8]. Previous studies have shown that 4-MU can inhibit the proliferation, migration, and invasion of multiple cancer cells in vitro and in vivo, including pancreatic, prostate, skin, esophageal, and liver cancer cells [9,10,11,12]. The anti-cancer effects of 4-MU treatment on the growth arrest and apoptosis of tumor cells is consistent with HA’s role in cell survival pathways.

There has been limited investigation of 4-MU effects on inflammation and autoimmunity. McKallip et al. reported that 4-MU treatment prevented lung injury and reduced inflammatory cytokine levels of inflammatory cytokines in enterotoxin- and lipopolysaccharide (LPS)-mediated acute lung injury models [13,14]. 4-MU has also been reported to ameliorate autoimmune disease in mouse models, specifically a collagen-induced arthritis model and an experimental autoimmune encephalomyelitis model [15,16]. 4-MU has protective influences against non-infectious forms of inflammation, including inflammation in the kidney due to ischemia and reperfusion as well as cigarette smoke-induced inflammation in the airways [17,18]. Conversely, 4-MU may worsen some inflammatory reactions. For example, 4-MU treatment worsened atherosclerosis severity in ApoE-deficient mice fed a high-fat diet [10]. It is not yet known whether 4-MU can block IgE-induced MC activation.

The effects of 4-MU on MC activation and associated inflammatory responses, and the molecular mechanisms mediating such effects, need to be clarified before 4-MU can be explored as a potential clinical treatment for allergic diseases. The purpose of this study was to investigate the potential anti-allergic effects of 4-MU on IgE-mediated MC activation. MCs were stimulated with anti-dinitrophenol (DNP) IgE antibodies and DNP-human serum albumin (HSA) antigen (Ag) in in-vitro cell experiments. Cell degranulation was evaluated by detection of β-hexosaminidase and histamine release. To explore molecular mechanism, 4-MU effects on FcεRI-mediated signaling protein expression was analyzed. Finally, 4-MU effects on allergic responses were studied in two in-vivo models: IgE/Ag-stimulated passive cutaneous anaphylaxis (PCA) and ovalbumin (OVA)-induced active systemic anaphylaxis (ASA).

## 2. Materials and Methods

### 2.1. Reagents

The reagents 4-MU (CID: 5280567), ketotifen fumarate (CID: 5282408), Evans blue (CID: 9409), toluidine blue (CID: 7083), and NF-κB inhibitor BAY 11-7082 (CID: 5353431) were purchased from Meilun Biotechnology Co., Ltd. (Dalian, China). ERK inhibitor U0126 (CID: 3006531), JNK inhibitor SP600125 (CID: 8515), and p38 inhibitor SB203580 (CID: 176155) were purchased from MedChem Express (Monmouth Junction, NJ, USA). DNP-HSA, monoclonal DNP-specific IgE and OVA were obtained from Sigma-Aldrich (St. Louis, MO, USA). Aluminum adjuvant was purchased from Thermo Scientific (Waltham, MA, USA). Rabbit monoclonal IgG primary antibodies against JNK (c-Jun N-terminal kinase), phosphorylated (p)-JNK (Thr183/Tyr185), p-SYK (spleen tyrosine kinase, Tyr525/ Tyr526), ERK (extracellular signal-regulated kinase), p-ERK (Thr202/Tyr204), p38, p-p38 (Thr180/Tyr182), and anti-GAPDH (glyceraldehyde 3-phosphate dehydrogenase) (Cell Signaling Technology, Danvers, MA). We bought rabbit monoclonal primary antibodies targeting SYK, p65, and IκBα from Abcam (Cambridge, MA) and bought IgGκ BP-horse radish peroxidase (HRP), anti-rabbit IgG-HRP, and monoclonal IgG anti-p-p65 (Ser311) primary antibodies from Santa Cruz Biotechnology (Dallas, TX, USA). 

### 2.2. Cell Culture

Rat basophilic leukemia-2H3 cells (RBLs; Cellcook Biotech, Guangzhou, China) were cultured in complete DMEM (Dulbecco’s modified Eagle medium) supplemented with 4.0 mM L-glutamine, 1.0 mM sodium pyruvate, and 100 U/mL penicillin, as well as 100 µg/mL streptomycin, non-essential amino acids (Solarbio, Beijing, China), and 10% fetal bovine serum (Gibco, Grand Island, NY, USA). The cultures were maintained in a humidified incubator (37 °C, 5% CO_2_).

Mouse bone marrow derived mast cells (BMMCs) isolated from BALB/c mouse femurs were cultured in complete RPMI-1640 media supplemented with 100 U/mL penicillin, 100 μg/mL streptomycin, 2 mM L-glutamine, 1 mM sodium pyruvate, and 10 mM 4-(2-hydroxyethyl)-1-piperazineethanesulfonic acid (HEPES), as well as 10% fetal bovine serum, 10 ng/mL IL3, and 10 ng/mL stem cell factor (SCF). We used flow cytometry to detect CD117 and FcεRI on cell surfaces after the cells had been cultured for 4–6 weeks; the flow cytometry outputs indicated that we had obtained ≥95% MC purity [19]. 

### 2.3. Cytotoxicity Assay

Cytotoxicity of 4-MU was determined with a Cell Counting Kit 8 assay kit in accordance with instructions provided by the vendor (MedChem Express, Monmouth Junction, NJ, USA). RBLs (2 × 10^3^/well) and BMMCs (1 × 10^4^/well) were incubated in separate 96-well plates, each with a range of 4-MU concentrations for 24 h. After the cells were held in a working solution for an hour, we used a microplate reader (Bio-Rad, Hercules, CA, USA) to take absorbance readings at 450 nm.

### 2.4. β-Hexosaminidase Release Assay

Degranulation extent was determined by measuring β-hexosaminidase activity, as described previously [20]. Briefly, RBLs and BMMCs were incubated with 50 ng/mL anti-DNP-IgE in complete media overnight for sensitization. The IgE-sensitized cells were washed with phosphate buffered saline (PBS), pretreated with 4-MU in Tyrode’s buffer for 1 h, and then challenged with 100 ng/mL of DNP-HSA for 30 min at 37 °C.

Supernatants and cell pellets were solubilized in 0.1% Triton X-100/Tyrode’s buffer. Each solution was incubated at 37 °C with an equal volume of substrate solution (1 mM 4-nitrophenyl-*N*-acetyl-β-d-glucosaminide) in 0.1 M sodium citrate buffer (pH 4.5). After 90 min, the reaction was halted with 150 μL of stop solution consisting of a mixture of 0.1 M Na_2_CO_3_/NaHCO_3_). A product of β-hexosaminidase activity was measured by detecting 405-nm absorbance in a plate reader (Bio-Rad).

### 2.5. Reverse Transcriptase (RT)-Quantitative Polymerase Chain Reaction (qPCR)

Following a 1-h 4-MU treatment of IgE-sensitized BMMCs and RBLs that had previously been stimulated with DNP-HSA for 4 h, total RNA was extracted from the cells with an RNeasy Mini Kit (Qiagen, Duesseldorf, Germany) in accordance with the manufacturer’s instructions. Complementary DNA was synthesized from 1 μg of total RNA using the HiScript III RT SuperMix (Vazyme, Nanjing, China) according to the manufacturer’s instructions. Gene expression levels were analyzed by real-time RT-qPCR with a TB Green^®^ Premix ExTaq^TM^ (Takara, Tokyo, Japan) in a qTOWER 2.2 system (Analytik Jena, Upland, CA, USA). The following primer sequences were used for genes of interest in Appendix A. Gene expression levels of target genes were normalized relative to *Gapdh*.

### 2.6. Western Blotting

To determine the effect of 4-MU on the signaling molecules in IgE-mediated MC activation, RBLs were sensitized with 50 ng/mL of DNP-IgE overnight, treated with 4-MU (10 μM or 20 μM) for 1 h, challenged with 0.1 μg/mL DNP-HSA for 30 min, and then harvested and lysed in RIPA buffer containing protease inhibitor cocktail (MedChem Express) for 10 min at 4 °C. Protein concentrations were measured with a BCA protein kit (Beyotime, Beijing, China). Protein samples (20 μg/well) were loaded onto an SDS-PAGE gel, transferred to polyvinylidene difluoride membranes (Merck Millipore, Billerica, MA, USA), and blocked in 5% bovine serum albumin in tris-buffered saline with 0.1% Tween-20 (TBS-T). The membranes were incubated overnight at 4 °C with anti-SYK [1:1000], anti-p-SYK [1:1000], anti-ERK [1:1000], anti-p-ERK [1:1000], anti-JNK [1:1000], anti-p-JNK [1:1000], anti-p38 [1:1000], anti-p-p38 [1:1000], anti-p-p65 [1:200], anti-p65 [1:5000], anti-IκBα [1:2000] and anti-GAPDH [1:2000] primary antibodies in TBS-T. The membranes were then incubated for 1 h with HRP-conjugated secondary antibodies [1:2000]. Immunoreactive protein bands were visualized by reaction with chemiluminescent reagents (Meilun Biotechnology Co., Ltd.). The relative levels of p-SYK/SYK, p-ERK/ERK, p-JNK/JNK, p-p38/p38, p-p65/p65 and IκBα/GAPDH were analyzed statistically using Image J software (National Institutes of Health, Bethesda, MD, USA).

### 2.7. Toluidine Blue Staining

We stained RBLs with toluidine blue (1% *w*/*v* in 1% saline, pH 2.5) as described in detail elsewhere [21], and then used an inverted microscope (Carl Zeiss, Goettingen, Germany) to observe the presence of heterochromatin particles. The numbers of IgE-activated cells and non-activated cells were each counted in five randomized visual fields.

### 2.8. F-actin Staining

F-actin recombination characteristic of MC activation [22] was examined with fluorescein isothiocyanate (FITC)-labeled phalloidin staining. Briefly, stimulated RBLs were fixed with 4% paraformaldehyde, incubated in a 200-µL aliquot of fluorescein isothiocyanate (FITC)-labeled phalloidin (Yeasen, Shanghai, China) for 30 min. After F-actin staining, the cells were observed with a fluorescence microscope (Carl Zeiss, Jena, Germany).

### 2.9. IgE-Mediated PCA Mouse Model

An anti-DNP-specific IgE (500 ng/ear) was injected intradermally into mouse ears. Then, 24 h later, topical 4-MU (50 mg/kg) was applied to ears or ketotifen (50 mg/kg; positive control) was injected intraperitoneally (i.p.) (N = 5/group). An hour later, the mice were challenged with DNP-HSA (200 µg in PBS with 0.5% Evans blue dye, given by tail vein injection). The mice were euthanized 1 h later. The ear thickness was measured by electronic digital caliper (Deli, Zhejiang, China). The ears were removed and submerged in formamide (700 μL) at 65 °C for 12 h to allow dye extraction. Dye intensity was quantitated at 620 nm by a microplate reader (Thermo Fisher Scientific Inc., Waltham, MA, USA). Mouse ears were submerged in 4% formaldehyde and then embedded in paraffin in preparation for slicing. Finally, sections were submitted to toluidine blue staining [23].

### 2.10. OVA-Induced ASA Mouse Model

Mice (N = 5/group) were sensitized with OVA mixture solution (100 µg of OVA plus 2 mg of aluminum adjuvant in 200 µL of PBS) on days 0 and 7, as we have described previously [20,24]. 4-MU (50 mg/kg, i.p.) or ketotifen (50 mg/kg, i.p.) was injected on days 9, 11, and 13. On day 14, we injected OVA (10 mg/kg, i.p.) and then measured each mouse’s rectal temperature every 10 min for 90 min. The mice were then killed at the conclusion of the 90-min observation period. We collected blood samples from the orbital venous plexus and then measured histamine release in the blood samples with enzyme-linked immunoassay (ELISA) kits obtained from IBL (Hamburg, German) and measured cytokine IL4 release in the blood samples using a sandwich ELISA kit (Elabscience, Wuhan, China) in accordance with the manufacturer’s protocol.

### 2.11. Statistical Analyses

The data are expressed means from three independent experiments with standard deviations (SDs). One-way analyses of variance (ANOVAs) were conducted in Prism 8 (GraphPad, La Jolla, CA, USA) with a significance criterion of *p* < 0.05.

## 3. Results

### 3.1. 4-MU Suppresses FcεRI-Mediated Degranulation in RBLs

Cytotoxic screening confirmed that 4-MU (0.1–100 μM) does not reduce RBL viability significantly (Figure 1b). Thus, 20 μM and 40 μM doses were used for subsequent in vitro experiments. At these doses, 4-MU inhibited β-hexosaminidase and histamine release dose-dependently in IgE/Ag-mediated RBLs (Figure 1c,d), demonstrating a suppression of MC degranulation. RT-PCR assays showed decreased *IL1β*, *IL4*, and *TNFα* transcription in IgE-activated RBLs following 4-MU treatment (Figure 1e–g).

### 3.2. 4-MU Suppresses FcεRI-Mediated Morphological Changes in RBLs

IgE-activated RBLs were identified metachromatically by the dominant presence of distinctive secretory granules in the cytoplasm. Non-activated RBLs were elongated and contained purple-colored intra-cellular particles. IgE-stimulated RBLs had irregular shapes with less staining of secretory granules in the cytoplasm due to extensive degranulation. 4-MU inhibited these changes and FcεRI-signaling induced particle release in activated RBLs (Figure 2a,b). Additionally, F-actin staining revealed that non-activated RBLs had uniformly distributed and spindle-shaped F-actin molecules. Conversely, activated RBLs took on an elliptical shape that fit their F-actin cytoskeletal changes. Treatment with 4-MU prior to IgE stimulation inhibited these activation-associated changes in cytomorphology and F-actin cytoskeleton decomposition in activated RBLs (Figure 2c,d).

### 3.3. 4-MU Suppresses Phosphorylation of FcεRI-Linked Signaling Proteins in RBLs

Because crosslinking of Ag-bound IgE and FcεRI leads to MC degranulation via activation of the SYK, mitogen-activated protein kinase (MAPK), and nuclear factor-kappa B (NF-κB) signaling, we investigated the effects of 4-MU on SYK, MAPK, and NF-κB pathway activation in RBLs. Western blot analysis showed that pretreatment with 4-MU attenuated the phosphorylation of SYK, p38, ERK, JNK, and NF-κB p65 and increased IκBα levels in FcεRI-activated RBLs dose-dependently (Figure 3a–c). Similarly, down-regulation of MAPK or NF-κB signal activity by inhibitors (ERK inhibitor U0126, JNK inhibitor SP600125, p38 inhibitor SB203580, or NF-κB inhibitor BAY 11-7082) coupled with 4-MU treatment in RBLs did not synergistically inhibit IgE-mediated MC activation, as evidenced by β-hexosaminidase release (Figure 3d,e).

### 3.4. 4-MU Has an Inhibitory Effect on IgE/Ag-Stimulated Primary BMMCs

4-MU at concentrations up to 100 μM did not have significant effects on primary BMMC viability (Figure 4a), indicating it was not cytotoxic. 4-MU inhibited β-hexosaminidase release and histamine release from IgE/Ag-activated BMMCs in a dose-dependent manner (Figure 4b,c). RT-PCR showed that 4-MU attenuated IgE-induced increases in pro-inflammatory cytokine (*IL1β*, *IL4*, and *IL6*) expression of BMMCs (Figure 4d–f).

### 3.5. 4-MU Attenuated IgE-Mediated Allergic Reactions in PCA Mice

IgE/Ag induced PCA, confirming establishment of the model (Figure 5a). The ears of PCA model mice treated with 4-MU had markedly less dye diffusion (Figure 5a), dye extrusion (Figure 5b), thickness (Figure 5c), and edema (Figure 5d) than ears in the activated group not treated with 4-MU. These results show that 4-MU treatment diminished allergy responses in PCA mouse ears

### 3.6. 4-MU Attenuated Allergic Reactions in OVA-Induced ASA Mice

Following an OVA challenge in ASA model mice, established as shown in Figure 6a, rectal temperatures decreased over a 20–50-min period and an increase in histamine levels was observed. The typical ASA decrease in the rectal temperature were attenuated by injection of 50 mg/kg (i.p.) 4-MU or the positive control treatment with the anti-histamine drug ketotifen (Figure 6b). Concomitantly, those increases in serum histamine levels after the OVA challenge in ASA mice were also suppressed by 4-MU (Figure 6c). Furthermore, the 4-MU treatment reduced serum IL4 levels associated with allergic inflammation in OVA-challenged animals (Figure 6d).

## 4. Discussion

In the present study, 4-MU was shown to inhibit FcεRI-mediated MC degranulation, as evidenced by reduced release of histamine and β-hexosaminidase, decreased expression of inflammatory cytokines, and attenuation of morphological changes induced by DNP-IgE/HSA-stimulation. Notably, 4-MU inhibited the phosphorylation of FcεRI-mediated signaling proteins that have previously been shown to be associated with MC activation, including SYK, MAPKs, JNK, p38, and NF-κB pathway components. In our in-vivo experiments, 4-MU attenuated the DNP-IgE/HSA-induced PCA reaction dose-dependently and suppressed ASA responses.

Targeting MCs is an important approach for treating IgE-mediated allergic diseases. MC activation can be inhibited by various clinical therapeutic agents, including antihistamines, glucocorticoids, MC stabilizers, and leukotriene antibody antagonists [25,26,27,28,29]. Antihistamines prevent histamine molecules from binding H1 receptors, which then prevents the release of histamine [26]. Although second-generation antihistamines have fewer adverse secondary effects than their predecessors, they still have a sedating side effect and may thus depress psychomotor functions and interfere with cognitive functions, which can impede academic performance [30]. Although intranasal corticosteroid treatments can reduce inflammation by way of their regulatory effects on mediator release [31], long-term use of corticosteroid medication increases one’s risks of osteoporosis, bone fractures, cataracts, hyperglycemia, slow healing, infection, and headache [32,33]. In recent decades, there has been a resurgence in the use of natural medicinal products, including for the treatment of immune-related diseases [34,35]. The present findings showed that the coumarin derivative 4-MU can suppress IgE-mediated MC activation in vitro and in vivo.

There have been limited prior investigations into the effects of 4-MU on inflammatory diseases in several cell types. For example, 4-MU has been reported to reduce LPS-stimulated upregulation of inflammatory cytokines (IL-1, IL-6, IL-8, and TNFα) in corneal fibroblasts [36], LPS-induced cytokine (IL-1, IL-6, and TNFα) production in spleen cells [14], and prostaglandin synthesis in LPS-mediated astrocyte inflammatory reactions [37]. The present data indicating that 4-MU has anti-inflammatory effects in the course of IgE stimulated MC activation—decreasing the release of histamine and β-hexosaminidase and also decreasing expression of inflammatory cytokines induced by DNP-IgE/HSA-stimulation of MCs—extend the limited literature on the anti-inflammatory effects of 4-MU. Notably, our results are consistent with previous reports showing that tozasertib or PF-431396 can inhibit MC degranulation [24,38]. Together, these findings support the notion that 4-MU inhibits FcεRI-mediated MC inflammatory responses.

Previously, SYK, MAPK, and NF-κB pathways have been suggested to be involved in mediating signaling downstream of FcεRI activation that promotes MC degranulation [39,40]. Mechanistically, our study shows that 4-MU reduces activation of SYK, JNK, the NF-κB heterodimer component p65, and the MAPKs p38 and ERK, as evidenced by phosphorylated kinase levels, without reducing total expression of these kinases. The inhibitory effects of 4-MU on FcεRI-mediated signaling observed here were similar to previously reported effects of the MC activation inhibitor T-5224 [41]. Previously, 4-MU has also been shown to reduce JNK phosphorylation in IL-1-stimulated chondrocytes and LPS-mediated astrocytes [37,42], to enable inhibition of ERK phosphorylation in malignant pleural mesothelioma cells [43] and esophageal squamous cells [44], to inhibit NF-κB signaling in prostate cancer cells [45], and to increase in p-p38 levels in K562 chronic myelogenous leukemia cells [46]. Additionally, we found that 4-MU did not affect transcription of several FcεRI receptor genes, including *Fcer1A*, *Ms4a2*, and *Fcer1G* (Appendix A), suggesting that 4-MU may not influence IgE-FcεRI binding.

It has not been clarified how or whether 4-MU inhibitory effects on HA synthesis relate to the anti-inflammatory actions 4-MU. Moreover, the molecular mechanisms of HA regulation in FcεRI-mediated MC degranulation have yet to be substantively clarified. Interestingly, our findings suggest that HA production may be associated with FcεRI-mediated activation of MCs. If so, HA synthetase could represent an important target for MC activation.

4-MU is sold over-the-counter as a spasmolytic in several European countries [37]. Exploration of the pharmacodynamic effects of 4-MU in multiple disease models is needed to explore its potential for drug repurposing [7]. Notably, 4-MU prevented lung injury in two mouse models (staphylococcal enterotoxin- and LPS-mediated acute lung injury) [13,14]. Furthermore, 4-MU protected against non-infectious inflammation in renal ischemia-reperfusion injury model mice [17], in atherosclerosis model mice [10], and the high-fat diet-induced hyperglycemia model [47]. Our study extends these findings by showing a new anti-allergic effect of 4-MU in two models of IgE-mediated inflammation, namely the IgE/Ag-induced allergic reaction in PCA model mice and the OVA-stimulation induced effects on body temperature and serum histamine levels in ASA model mice.

A typical adult 4-MU dosage range is a total of 900–2400 mg/day, divided into three daily doses [7]. Previously, 4-MU has been delivered orally to arthritic mice at a dosage of 10–60 mg/kg per day for a period of 19 days [15]. Thus, the presently used 4-MU dose (50 mg/kg) was of a similar dosage as that used previously in mice and by no means excessive for clinical application.

## 5. Conclusions

In summary, we found that the FDA-approved plant-derived HA synthesis inhibitor 4-MU attenuates FcεRI-mediated activation of MCs and MC inflammatory responses. Our data suggest that the effects of 4-MU on inflammatory responses may involve MAPK-, SYK-, and/or NF-κB-dependent mechanisms. The presently demonstrated in vitro and in vivo effects of 4-MU on MC activation suggest that it should be considered a candidate of interest for being re-purposed as an MC inhibitor for allergic disease treatment.

## Figures and Tables

**Figure 1 molecules-27-01577-f001:**
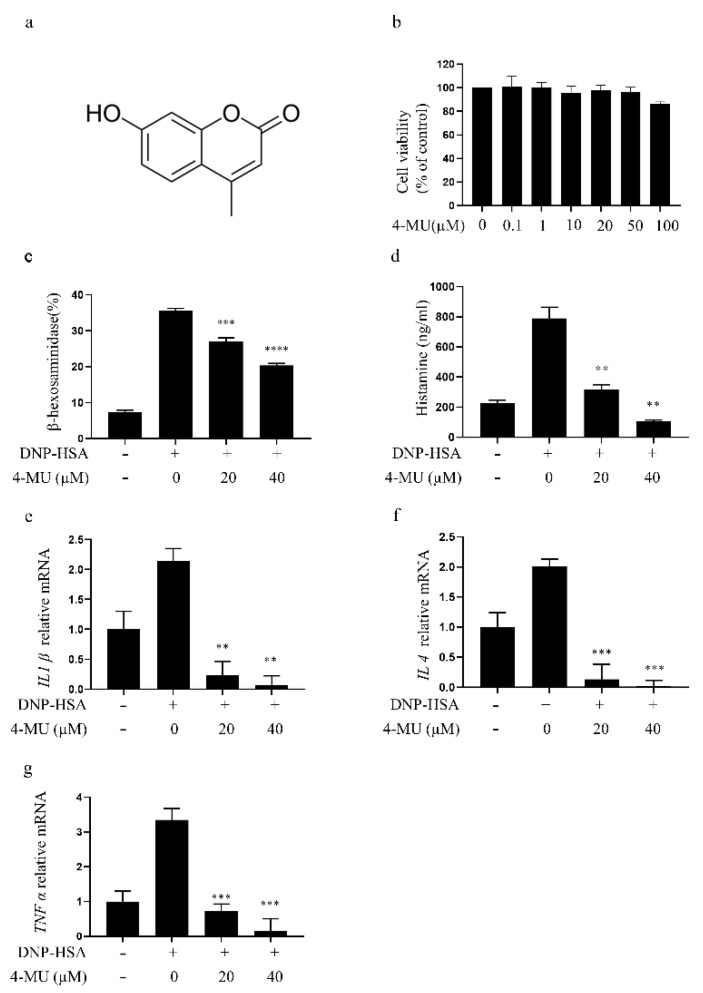
Treatment with 4-MU suppresses FcεRI-mediated degranulation in RBLs. (**a**) Chemical structure of 4-MU. (**b**) Viability of RBLs incubated with 0–100 μM 4-MU for 24 h determined by CCK-8 assays. (**c**–**g**) Secretion of β-hexosaminidase (**c**), histamine release (**d**), and expression of inflammatory cytokine genes *IL1β* (**e**), *IL4* (**f**), and *TNFα* (**g**) in RBLs sensitized with anti-DNP IgE, with or without 4-MU, for 1 h, and then challenged with DNP-HSA. The data are shown in means ± SDs (five duplicate experiments); ** *p* < 0.01, *** *p* < 0.001, **** *p* < 0.0001 vs. non-treated activated cells. 4-MU, 4-methylumbelliferone; DNP-HSA, dinitrophenyl-human serum albumin protein conjugate; RBL, RBL-2H3 cell.

**Figure 2 molecules-27-01577-f002:**
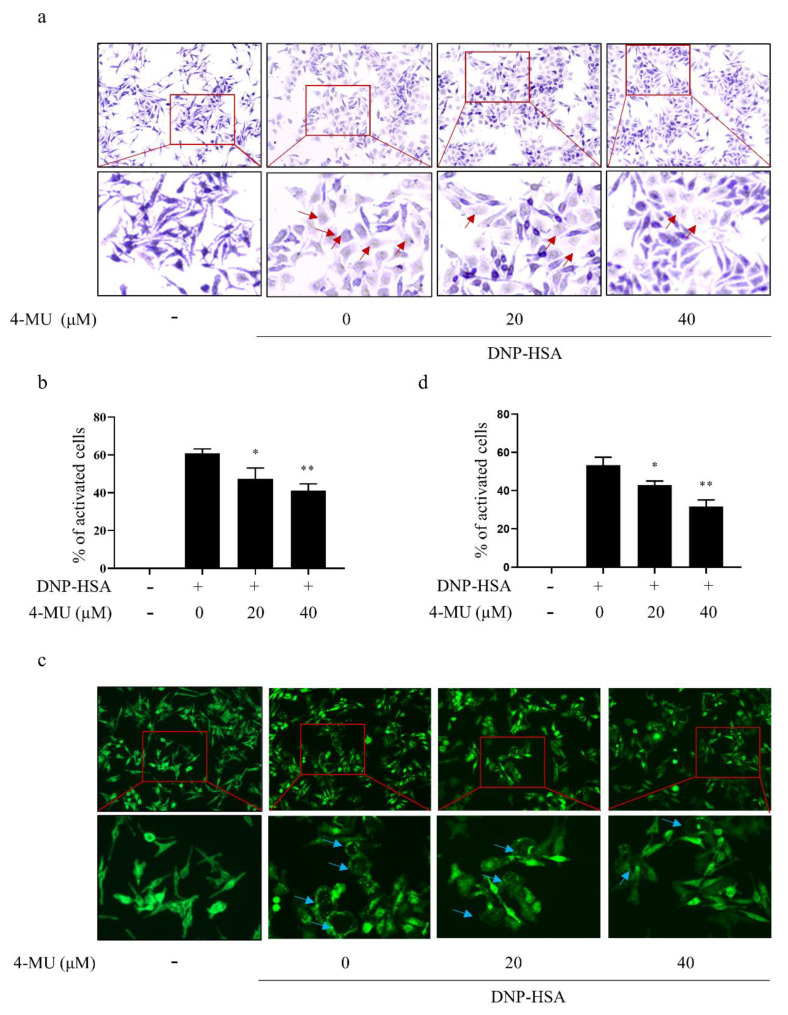
4-MU suppresses FcεRI-mediated morphological changes in RBLs. Anti-DNP IgE-sensitized RBLs were pretreated (or not) with 4-MU for 1 h and then challenged with DNP-HSA (100 ng/mL) for 30 min. (**a**,**b**) Representative micrographs of toluidine blue-stained RBLs. Red arrows indicate irregular cell morphology and the release purple particles. (**c**,**d**) FITC-phalloidin stained RBLs. Blue arrow indicates cell morphology irregularity due to decomposition of F-actin cytoskeleton. Means ± SDs (of three independent experiments) are shown; * *p* < 0.05, ** *p* < 0.01 vs. activated cells without treatment. Abbreviations as in Figure 1.

**Figure 3 molecules-27-01577-f003:**
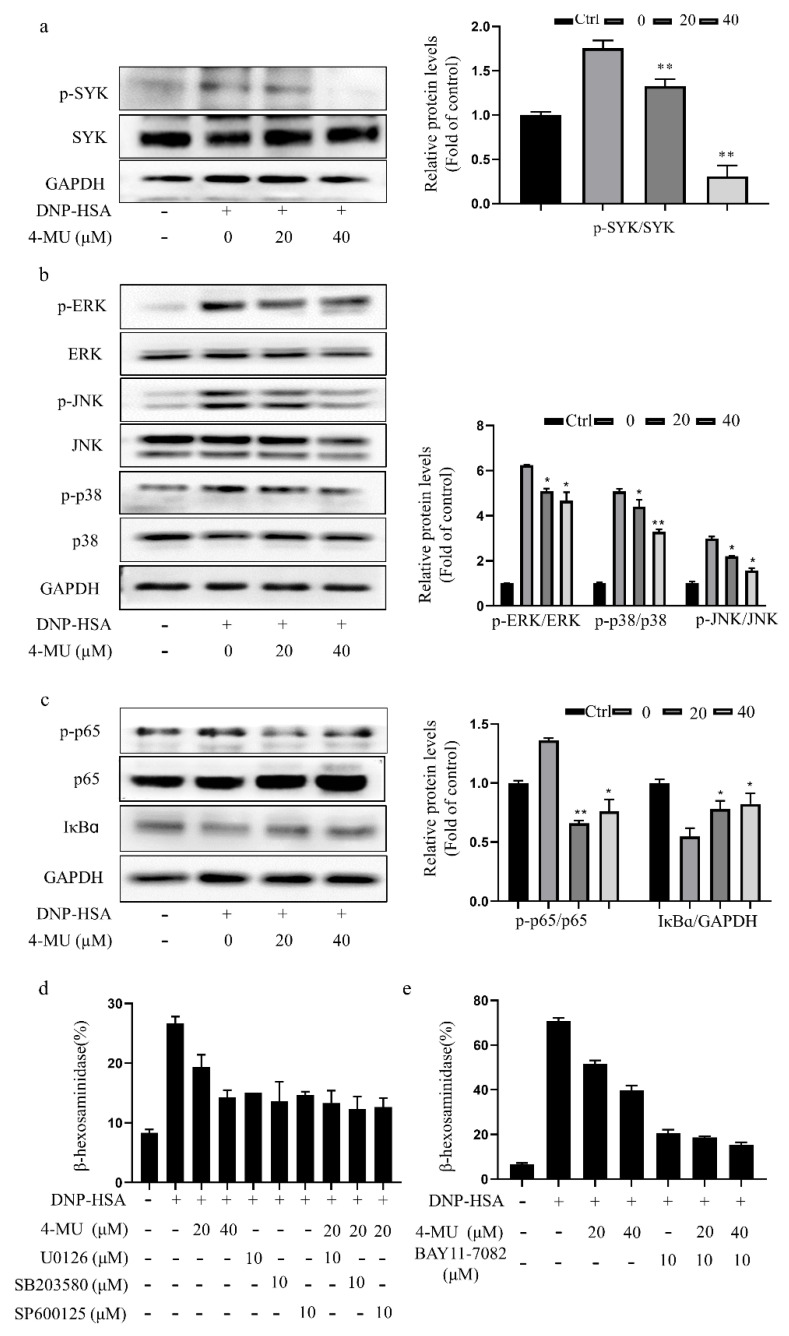
4-MU suppresses FcεRI-mediated signaling protein activation in RBLs. IgE-sensitized RBLs were pre-incubated with 4-MU for 1 h and then stimulated with DNP-HSA for 20 min. (**a**–**c**) Western blot analysis of SYK (**a**), MAPK (p38, ERK, and JNK) (**b**), and NF-κB pathway signaling molecules (p-p65, p65, and IκBα) (**c**) in activated RBLs. (**d**,**e**) β-Hexosaminidase release from IgE-activated RBLs treated with 4-MU and MAPK inhibitors (ERK inhibitor U0126, JNK inhibitor SP600125, and p38 inhibitor SB203580) (**d**) or the NF-κB inhibitor BAY 11-7082 (**e**) for 1 h prior to the DNP-HSA challenge. Means (of three duplicate experiments) ± SDs are shown; * *p* < 0.05, ** *p* < 0.01 vs. activated cells without treatment. Abbreviations as in Figure 1.

**Figure 4 molecules-27-01577-f004:**
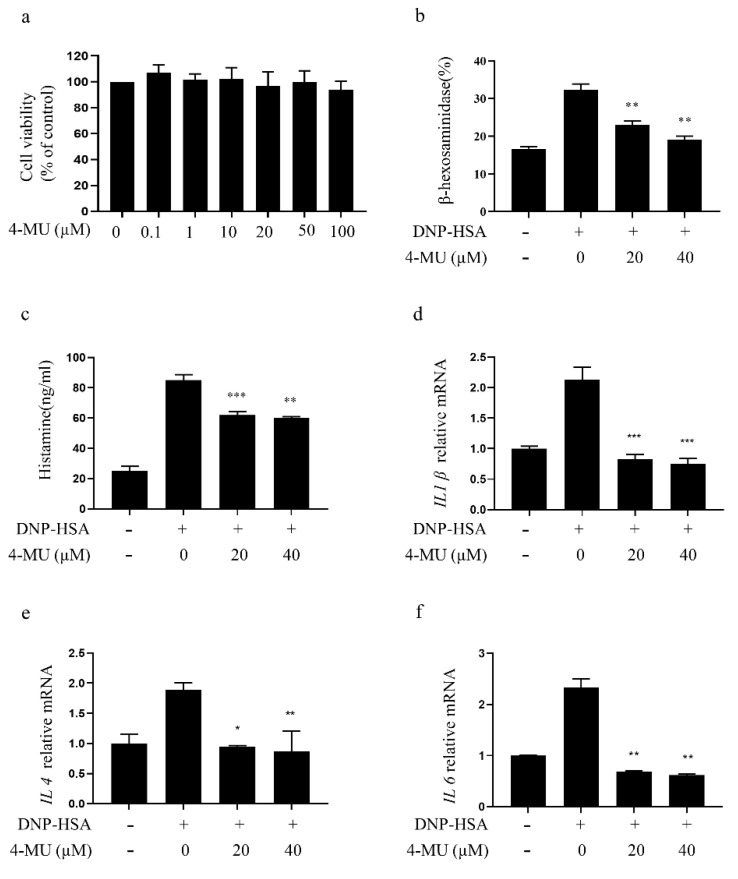
4-MU inhibition of IgE/Ag stimulation of primary BMMCs. (**a**) Cell viability, as determined by CCK-8 assay, of BMMCs incubated with 0–100 μM of 4-MU for 24 h. Cells were sensitized with anti-DNP IgE, with or without 4-MU, for 1 h, and then challenged with DNP-HSA. (**b**) β-Hexosaminidase release. (**c**) Histamine release. (**d**–**f**) mRNA expression of *IL1β* (**d**), *IL4* (**e**), and *IL6* (**f**) determined by RT-qPCR. BMMCs were sensitized with anti-DNP IgE with or without 4-MU for 1 h and challenged with DNP-HSA for 4 h. Means ± SDs (of five duplicate experiments) are shown; * *p* < 0.05, ** *p* < 0.01, *** *p* < 0.001. Abbreviations as in Figure 1.

**Figure 5 molecules-27-01577-f005:**
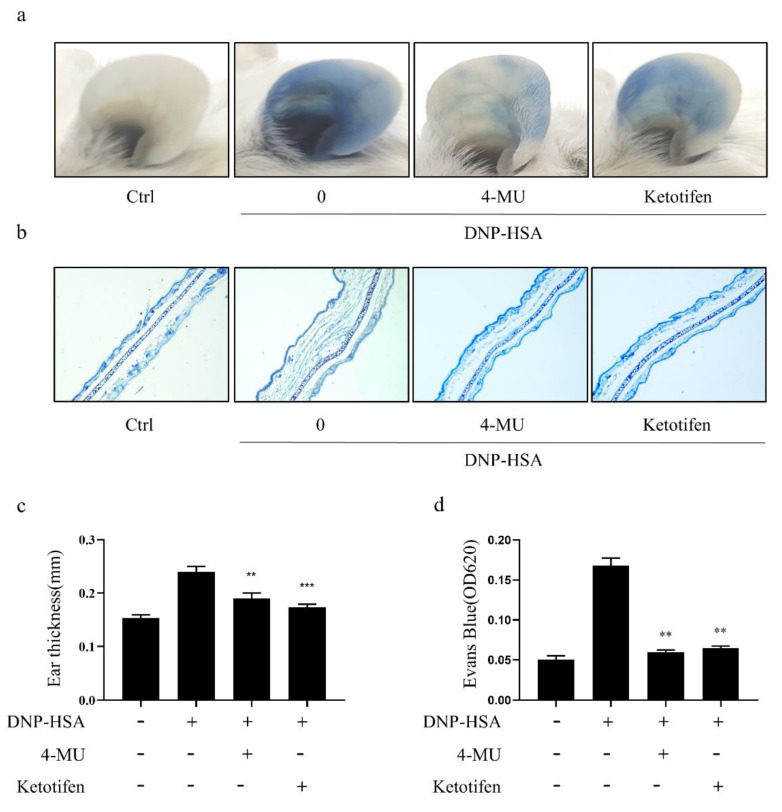
4-MU attenuation of IgE-mediated allergic reactions in PCA mice [sensitized with anti-DNP-IgE (or saline control) for 12 h, 4-MU for 1 h, and injected with DNP-HSA/0.5% Evans blue]. (**a**) Representative photos of showing dye extravasation from ears. (**b**) Representative photomicrographs of sections from PCA ears. (**c**) Reversal of DNP-HSA-induced increases in ear thickness. (**d**) Formaldehyde-extracted Evans blue dye quantified at 620 nm. Means (of three independent experiments) ± SD are shown; ** *p* < 0.01, *** *p* < 0.001 vs. activated cells, not treated. Abbreviations as in Figure 1.

**Figure 6 molecules-27-01577-f006:**
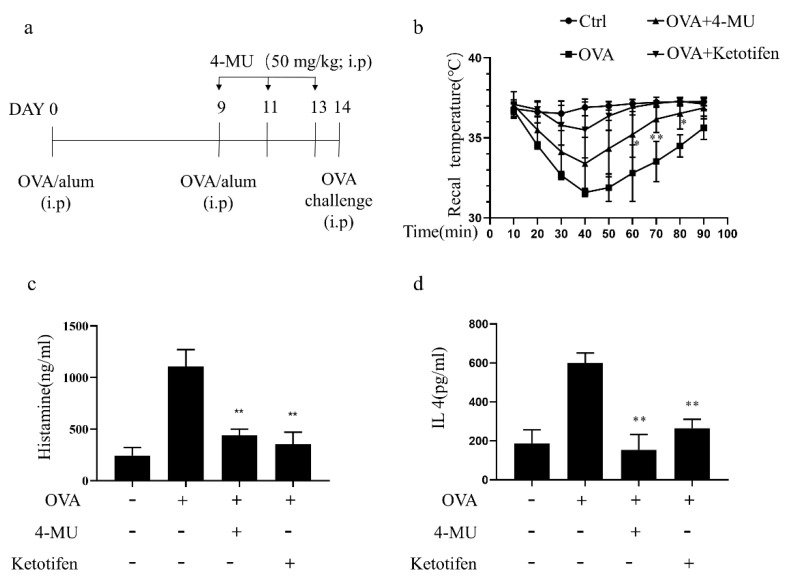
4-MU attenuates allergic reactions in ASA mice. (**a**) ASA protocol (ketotifen, positive control). (**b**) Mean (± SD) rectal temperatures taken every 10 min after initiation of an OVA challenge (three independent experiments). (**c**) Effects of 4-MU on histamine release into serum (determined by ELISA). (**d**) Effects of 4-MU on serum IL-4 levels in ASA mice (determined by ELISA). Means ± SDs (three independent experiments) are shown (N = 5/group); * *p* < 0.05, ** *p* < 0.01 vs. activated group not pretreated with 4-MU.

## Data Availability

The data presented in this study are available upon request from the corresponding authors.

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
