# Peer review of "Plant-Derived Molecule 4-Methylumbelliferone Suppresses FcεRI-Mediated Mast Cell Activation and Allergic Inflammation"

_molecules, 2022, doi:10.3390/molecules27051577_

Round 1
Reviewer 1 Report
Authors investigated the effects of 4-methylumbelliferone (4-MU) on the mast cell (MU) degranulation and its mechanism. Rat basophilic leukemia cells (RBLs) and mouse bone marrow-derived mast cells (BMMCs) were used for their in vitro studies. 4-MU suppressed FceRI-mediated degranulation as well as cytomorphological elongation and F-actin reorganization in RBLs and BMECs.
Furthermore, the IgE/Ag-induced phosphorylation of SYK, NF-kB p65, ERK1/2, p38, and JNK was reduced by 4-MU. In vivo, 4-MU attenuated allergic reactions as shown by using the passive cutaneous anaphylaxis (PCA) and the active systemic anaphylactic (ASA) murine models. Authors demonstrated that 4-MU reduced dye extravasation and ear thickness (PCA) as well as body temperature and serum histamine levels (ASA). Authors concluded from these results that 4-MU may represent a new candidate for the treatment of allergic diseases by suppressing MC activation.
These studies complete the knowledge about anti-inflammatory activities of 4-MU. However, the relevance of this study should be improved even more.
Major points:
Authors demonstrated that degranulation of mast cells and signaling pathways are changed by 4-MU. However, they have to confirm that the binding between IgE and FceRI and also the expression of the receptor on the cell surface is not changed by 4-MU.
Authors should further clarify whether the tyrosine phosphorylation of PLCg and the increase in intracellular Ca2+ is modulated by 4-MU. The role of the intracellular Ca2+ increase can be also performed by using other mast cell activators.
Figure 1, showing the FceRI-mediated degranulation in RBLs, should be completed by adding the histamine release and the cytokine expression as shown in Fig. 4 for BMMCs.
The images of Figure 2a and 2c demonstrating the toluidine- and actin staining should be enlarged to see more details and furthermore the methods of their quantification should be described.
The in vivo experiments (Figure5 and Figure 6) should be amended by an additional control which inhibited the degranulation of mast cells such as cromolyn.
Minor points:
Figure 1 should be put to another position
Legends of Fig.3 and Fig.4 have to be inverted.
Author Response
Questions and Answer:
- Authors demonstrated that degranulation of mast cells and signaling pathways are changed by 4-MU. However, they have to confirm that the binding between IgE and FceRI and also the expression of the receptor on the cell surface is not changed by 4-MU.
Reply: Thank you for your constructive suggestion. Real-time RT-PCRs were performed to detect the expression of the FcεRI receptors including FcεRIα, FcεRIβ and FcεRIγ.The following sequence-specific forward (F) and reverse (R) primers were used (5’ to 3’): rat Fcer1A (F, GGCTGCTGCTCCAATCTTC; R, GCAATGTCGTCCTTGTAGTAGA), rat Ms4a2 (F, TGCTCCACACTCCAGACTTC; R, GCTGCCTCTCACCAGATACA), rat Fcer1G (F, GGTGATCTTGTTCTTGCTCCTT; R, TCACGGCTGGCTATGTCTG), mouse Fcer1A (F, CCGTCTCTGAGGTGAACTCTT; R, CAGCCAATCTTGCGTTACATTC, mouse Ms4a2 (F, TGTAGAAGTCTGGATGGTGGAA; R, CTAAGTGTAGGCATGTGGAGTT) and mouse Fcer1G (F, TGCTTTGAAGGTTGGCTGAC; R, AAGGAGGCTGGAAGAAGAGAA). The result showed that the expression of the FcεRI receptors on the mast cell surface is not changed by 4-MU treatment (Supplemental Figure 1).
The Supplemental Figure 1:
- Authors should further clarify whether the tyrosine phosphorylation of PLCg and the increase in intracellular Ca2+ is modulated by 4-MU. The role of the intracellular Ca2+ increase can be also performed by using other mast cell activators.
Reply: It is a good advice. The focus of this manuscript was to study the anti-allergic effect of 4-MU on IgE-mediated mast cells activation. According to previous literatures, IgE-FcεRI crosslinking leads to the activation of downstream pathways, including mitogen-activated protein kinase (MAPK), Ca2+, and nuclear factor kappa-B (NF‐κB) signaling pathways that function to regulate de novo synthesis of lipid mediators and cytokines involved in the development of allergic inflammation [Eur J Pharmacol 778 (2016) 158-68.]. In our manuscript, we found that 4-MU inhibited the IgE-induced SYK phosphorylation, which can increase the intracellular calcium influx in mast cells. Additionally, pretreatment with 4-MU attenuated the phosphorylation of p38, ERK, JNK, and NF-κB p65 and increased IκBα levels in FcεRI-activated RBLs dose-dependently (Figure 3). These results indicate that 4-MU may suppress activation of IgE/Ag-stimulated MCs through down-regulation of MAPK and NF-κB pathways. Based on these results, it can confirm that 4-MU can suppress the key molecular pathway of mast cells activation.
The manuscript mainly pay attention to anti-allergic effect of 4-MU on IgE-mediated mast cells activation, especially effective factors. In the revised Figure 1, pretreatment with 4-MU attenuated the IgE-induced increases in the expression of pro-inflammatory factors by RBL-2H3 cells, including IL1β, IL4, and Tnfα (Figure 1). Hope you can also agree with our answer.
The revised Figure 1:
- Figure 1, showing the FceRI-mediated degranulation in RBLs, should be completed by adding the histamine release and the cytokine expression as shown in Fig. 4 for BMMCs.
Reply: Thank you for your constructive suggestion. In the revised Figure 1, 4-MU inhibited β-hexosaminidase and histamine release dose-dependently in IgE/Ag-mediated RBLs (Figure 1c and d), demonstrating a suppression of MC degranulation. In additional, our results of RT-PCR assays showed decreased IL1β, IL4 and TNFα transcriptional expression in IgE-activated RBLs following 4-MU treatment (Fig. 1e-g).
- The images of Figure 2a and 2c demonstrating the toluidine- and actin staining should be enlarged to see more details and furthermore the methods of their quantification should be described.
Reply: Thank you for your constructive suggestion. We have revised the related section.
The revised section in the methods:
“The number of IgE-activated cells and non-activated cells was counted in five randomized visual fields, respectively.”
The revised Figure 2:
- The in vivo experiments (Figure5 and Figure 6) should be amended by an additional control which inhibited the degranulation of mast cells such as cromolyn.
Reply: Ketotifen, an anti-histamine drug, was well-known as a classical mast cell stabilizer in treating the mast cells activation induced allergic inflammation [FASEB J . 2020; 34(8): 10117–10131]. As shown in Figure 5 and Figure 6, the ketotifen was used as positive control treatment group in the in vivo experiments. This practice is recognized by most of the peers, and the ketotifen control group was also used in our previous studies [Biochem Pharmacol. 2021 Oct;192:114722; J Transl Med. 2021 Jun 15;19(1):261; Int Immunopharmacol. 2020 Jul;84:106500; Br J Pharmacol. 2020 Jun;177(12):2848-2859].
- Figure 1 should be put to another position. Legends of Fig.3 and Fig.4 have to be inverted.
Reply: Thanks for your suggestion. We have agreed with you. We have revised them.

Reviewer 2 Report
Based on the idea, applied methods, as well as the obtained result, the paper meets all technical and scientific criteria. Before considering acceptance, it is necessary to check the English language, as well as check for technical errors throughout the text. The conclusion should be expanded in more detail on the basis of all the results obtained. It is necessary to discuss the scientific novelty well in the discussion, as well as to emphasize the application of the results in more detail.
Author Response
Reviewer 2
Based on the idea, applied methods, as well as the obtained result, the paper meets all technical and scientific criteria. Before considering acceptance, it is necessary to check the English language, as well as check for technical errors throughout the text. The conclusion should be expanded in more detail on the basis of all the results obtained. It is necessary to discuss the scientific novelty well in the discussion, as well as to emphasize the application of the results in more detail.
Reply: Thank you for your agreement. We have revised some points according to your and other reviewers’ professional suggestions.
- We found that the transcriptional level of the FcεRI receptor genes including Fcer1A, Ms4a2 and Fcer1G were not changed by 4-MU treatment (Supplemental Figure 1), indicating that 4-MU may not influence the binding between IgE and FcεRI receptors. Real-time RT-PCRs were performed to detect the expression of the FcεRI receptors including FcεRIα, FcεRIβ and FcεRIγ. The following sequence-specific forward (F) and reverse (R) primers were used (5’ to 3’): rat Fcer1A (F, GGCTGCTGCTCCAATCTTC; R, GCAATGTCGTCCTTGTAGTAGA), rat Ms4a2 (F, TGCTCCACACTCCAGACTTC; R, GCTGCCTCTCACCAGATACA), rat Fcer1G (F, GGTGATCTTGTTCTTGCTCCTT; R, TCACGGCTGGCTATGTCTG), mouse Fcer1A (F, CCGTCTCTGAGGTGAACTCTT; R, CAGCCAATCTTGCGTTACATTC, mouse Ms4a2 (F, TGTAGAAGTCTGGATGGTGGAA; R, CTAAGTGTAGGCATGTGGAGTT) and mouse Fcer1G (F, TGCTTTGAAGGTTGGCTGAC; R, AAGGAGGCTGGAAGAAGAGAA). The result showed that the expression of the FcεRI receptors on the mast cell surface is not changed by 4-MU treatment (Supplemental Figure 1).
- In the revised Figure 1, pretreatment with 4-MU attenuated the IgE-induced increases in the expression of pro-inflammatory factors by RBL-2H3 cells, including IL-1β, IL-4, and TNFα.
- In the revised Figure 6, the 4-MU treatment reduced serum IL4 levels associated with allergic inflammation in OVA-challenged animals (6D).
- Due to our unclear expression, We have revised it in the results of Figure 6. “Following an OVA challenge in ASA model mice, established as shown in Figure 6A, rectal temperatures decreased over a 20–50-min period and an increase in histamine levels was observed. The typical ASA decrease in the rectal temperature were attenuated by injection of 50 mg/kg (i.p.) 4-MU or the positive control treatment with the anti-histamine drug ketotifen (Figure 6B). Concomitantly, those increases in serum histamine levels after the OVA challenge in ASA mice were also suppressed by 4-MU (Figure 6C). ”
Hope you could agree with our revisions.

Reviewer 3 Report
The manuscript entitled “Plant-derived molecule 4‐methylumbelliferone suppresses FcεRI-mediated mast cell activation and allergic inflammation” by Hui-Na Wang and coworkers have focused on understanding the effect of plant-derived molecule 4-methylumbelliferone (4-MU) on mast cells degranulation and the underlying molecular mechanism. Using in vivo and in vitro approaches the authors have demonstrated that 4-MU by downregulating the IgE/Ag-induced SYK, NF-κB p65, ERK1/2, p38, and JNK phosphorylation, attenuate the mast cells degranulation and can be used as a treatment strategy to treat IgE-mediated allergic diseases. The findings of the manuscript are interesting. I have some concerns:
- Figure 1: panel D, E & F, and the associated results are missing. The authors should check the figure and result section.
- Figure 2 A and C: The authors should add some high magnification images to demonstrate morphological changes and the difference in F actin staining between the groups.
- Figures 3 and 4: The figure legend looks interchanged. The authors should check it carefully.
- Figure 4: IL-27 signaling negatively regulates mast cell activation and its mediated allergic response. Did the authors check the effect of 4-MU on IL-27 expression (in vivo/in vitro)?
- Figure 5B: The difference in Evans blue dye extrusion between the groups is not very clear. The authors should use more representative images.
- Figure 6C: Did the authors estimate serum histamine in PCA mouse model or ASA mouse model? The authors should check the same. The authors should also check the conclusion of line 258 “drop in histamine levels following OVA exposure”.
- Whether inhibition of NF-κB p65, ERK1/2, p38, JNK singling synergizes the effect of 4-MU and suppress mast cells degranulation. Did the authors analyzed the effect of 5-MU on proinflammatory cytokines level under in vivo settings?
- Line 247: Figure 5C and D in the result section are mislabeled.
- The authors should extensively check the typographical errors throughout the manuscript.
Author Response
Questions and Answer:
1. Figure 1: panel D, E & F, and the associated results are missing. The authors should check the figure and result section.
Reply: Thanks for your reminding. We have revised the mistake. 4-MU inhibited β-hexosaminidase and histamine release dose-dependently in IgE/Ag-mediated RBLs (Figure 1c and d), demonstrating a suppression of MC degranulation. In additional, our results of RT-PCR assays showed decreased IL1β, IL4 and TNFα transcriptional expression in IgE-activated RBLs following 4-MU treatment (Fig. 1e-g).
The revised Figure 1:
2. Figure 2 A and C: The authors should add some high magnification images to demonstrate morphological changes and the difference in F actin staining between the groups.
Reply: Thank you for your constructive suggestion. We have revised the related section.
The revised section in the methods:
“The number of IgE-activated cells and non-activated cells was counted in five randomized visual fields, respectively.”
The revised Figure 2:
3. Figures 3 and 4: The figure legend looks interchanged. The authors should check it carefully.
Reply: Thanks for your reminding. We have revised the related section.
4. Figure 4: IL-27 signaling negatively regulates mast cell activation and its mediated allergic response. Did the authors check the effect of 4-MU on IL-27 expression (in vivo/in vitro)?
Reply: We are grateful to your suggestion. It is found that L-27 signaling negatively regulates mast cell activation and its mediated allergic response [J Leukoc Biol. 2022 Jan 24. doi: 10.1002/JLB.2MA1221-637R]. However, the focus of this manuscript was to study the anti-allergic effect of 4-MU on IgE-mediated mast cells activation. It should be to test the cellular effectors, which can reflect the mast cells activation, especially inflammatory cytokines or mediators, as well as mast cells activation associated signal molecules. In the revised Figure 1, pretreatment with 4-MU attenuated the IgE-induced increases in the expression of pro-inflammatory factors by RBL-2H3 cells, including IL1β, IL4, and Tnfα (Figure 1).
Whether 4-MU can regulate the IL-27 expression in IgE-mediated mast cells is a valuable direction. Unluckily, we have found no transcriptional level change of IL-27 gene in 4-MU treated RBL-2H3 cells using RT-PCR method (Data not shown). The following sequence-specific forward (F) and reverse (R) primers were used (5’ to 3’): (F, TTCCTCGCTACCACACTT; R, TCCTCTTCTTCCTCTTCCTT). It indicated that 4-MU may not regulate the IL-27 expression in mast cells. Hope you can agree with our answer.
5. Figure 5B: The difference in Evans blue dye extrusion between the groups is not very clear. The authors should use more representative images.
Reply: Thanks for your reminding. We have revised the Fig.5B.
The revised Figure 5B:
6. Figure 6C: Did the authors estimate serum histamine in PCA mouse model or ASA mouse model? The authors should check the same. The authors should also check the conclusion of line 258 “drop in histamine levels following OVA exposure”.
Reply: Sorry to our unclear expression. Figure 6C showed the serum histamine in ASA mouse model. We have revised it in the manuscript. “Following an OVA challenge in ASA model mice, established as shown in Figure 6A, rectal temperatures decreased over a 20–50-min period and an increase in histamine levels was observed. The typical ASA decrease in the rectal temperature were attenuated by injection of 50 mg/kg (i.p.) 4-MU or the positive control treatment with the anti-histamine drug ketotifen (Figure 6B). Concomitantly, those increases in serum histamine levels after the OVA challenge in ASA mice were also suppressed by 4-MU (Figure 6C). Furthermore, the 4-MU treatment reduced serum IL4 levels associated with allergic inflammation in OVA-challenged animals (Fig. 6d).”
The serum histamine level is difficult to change in PCA model mice. On one hand, PCA model just generated local inflammation in the injected ear, not systemic inflammation. Local inflammation is also likely to increase the serum histamine level. But, since the model was challenged with an intravenous injection of 200-μl DNP-HSA (5 mg/ml Evans blue solution containing 0.1 mg/ml DNP-HSA), the Evans blue staining solution would impact the detection of serum histamine using the ELISA kit (IBL, German). Thus, the serum histamine from PCA models was not to be detected.
7. Whether inhibition of NF-κB p65, ERK1/2, p38, JNK singling synergizes the effect of 4-MU and suppress mast cells degranulation. Did the authors analyzed the effect of 5-MU on proinflammatory cytokines level under in vivo settings?
Reply: We thank for your suggestion. In Figure 3d and e, down-regulation of MAPK or NF-κB signal activity by molecule inhibitors (ERK inhibitor U0126, the JNK inhibitor SP600125, the p38 inhibitor SB203580, and the NF-κB inhibitor BAY 11-7082) coupled with 4-MU in RBLs did not synergistically inhibit IgE-mediated mast cell activation, as evidenced by detection of β-hexosaminidase release. In Figure 6d, the 4-MU treatment reduced serum IL4 levels associated with allergic inflammation in OVA-challenged animals (Fig. 6d).
The revised Figure 3d and e:
The revised Figure 6d:
8. Line 247: Figure 5C and D in the result section are mislabeled.
Reply: Thanks for your reminding. We have revised the related section.
The Revised section:
“4-MU attenuated allergy responses in PCA mouse ears, as evidenced by decreased dye diffusion (Figure 5a), diminished extrusion of dye from IgE/Ag-injected ears (Figure 5b), reduced ear thickness (Figure 5c) and decreased ear edema (Figure 5d), compared to the activated group not treated with 4-MU.”
9. The authors should extensively check the typographical errors throughout the manuscript.
Reply: We thank for your suggestion. We have revised some spells and done language editing in our manuscript by Professional English Language Editing Services (WRITE SCIENCE RIGHT, USA).

Round 2
Reviewer 1 Report
Authors responded to all points of the reviewer satisfactorily.
Author Response
Reply: Thank you for your agreement. Additionally, we have revised some spells and done language editing in our manuscript by Professional English Language Editing Services (WRITE SCIENCE RIGHT, USA).

Reviewer 3 Report
The authors have significantly modified the manuscript and answered most of related queries. However there are some minor corrections that need to be addressed before publication.
Minor correction:
Figure 1: the authors should include **p < 0.01 in the figure legend. In the result section (3.1) the authors mentioned use of 10 μM and 20 μM 4-MU doses, while the figure denotes 20 μM and 40 μM. The authors should check it.
Figure 5: The authors should remove *p < 0.05 and include ***p < 0.001 in figure legend.
Supplemental Figure 1: Figure legend is missing.
Method (Western blotting): The authors should include Ab for p65, p-p65, IkBa and GAPDH and also mention how the data was analyzed. The phosphorylation sites of all Abs used should be mentioned.
The method used to measure the ear thickness should be included.
Author Response
Reviewer 3
Questions and Answer:
The authors have significantly modified the manuscript and answered most of related queries. However there are some minor corrections that need to be addressed before publication.
Reply: We thank for your suggestion. We have revised some spells and done language editing in our manuscript by Professional English Language Editing Services (WRITE SCIENCE RIGHT, USA).
- Figure 1: the authors should include **p < 0.01 in the figure legend. In the result section (3.1) the authors mentioned use of 10 μM and 20 μM 4-MU doses, while the figure denotes 20 μM and 40 μM. The authors should check it.
Reply: Thanks for your reminding. We have revised the mistake. 20 μM and 40 μM doses is correct.
- Figure 5: The authors should remove *p < 0.05 and include ***p < 0.001 in figure legend.
Reply: Thanks for your reminding. OK.
- Supplemental Figure 1: Figure legend is missing.
Reply: In fact, the Figure legend is in our manuscript (doc or pdf). In the manuscript (LaTeX) developed by submission platform, Figure legend is missing. We have add it to the revised manuscript again.
- Method (Western blotting): The authors should include Ab for p65, p-p65, IkBa and GAPDH and also mention how the data was analyzed. The phosphorylation sites of all Abs used should be mentioned.
Reply: We have revised them. Thanks for your constructive suggestion.
Our revised manuscript:
“Rabbit monoclonal IgG primary antibodies against JNK (c-Jun N-terminal kinase), phosphorylated (p)-JNK (Thr183/Tyr185), p-SYK (spleen tyrosine kinase, Tyr525/ Tyr526), ERK (extracellular signal-regulated kinase), p-ERK (Thr202/Tyr204), p38, p-p38 (Thr180/Tyr182), and anti-GAPDH (glyceraldehyde 3-phosphate dehydrogenase) (Cell Signaling Technology, Danvers, MA). We bought rabbit monoclonal primary antibodies targeting SYK, p65, and IκBα from Abcam (Cambridge, MA) and bought IgGκ BP-horse radish peroxidase (HRP), anti-rabbit IgG-HRP, and monoclonal IgG anti-p-p65 (Ser311) primary antibodies from Santa Cruz Biotechnology (Dallas, TX). ”
“The membranes were incubated overnight at 4 °C with anti-SYK [1:1000], anti-p-SYK [1:1000], anti-ERK [1:1000], anti-p-ERK [1:1000], anti-JNK[1:1000], anti-p-JNK [1:1000], anti-p38 [1:1000], anti-p-p38 [1:1000], anti-p-p65 [1:200], anti-p65 [1:5000], anti-IκBα [1:2000] and anti-GAPDH [1:2000] primary antibodies in TBS-T. The membranes were then incubated for 1 h with HRP-conjugated secondary antibodies [1:2000]. Immunoreactive protein bands were visualized by reaction with chemiluminescent reagents (Meilun Biotechnology Co., Ltd (Dalian, China). The relative levels of p-SYK/SYK, p-ERK/ERK, p-JNK/JNK, p-p38/p38, p-p65/p65 and IκBα/GAPDH were analyzed statistically using Image J software (National Institutes of Health, Bethesda, MD, USA).”
- The method used to measure the ear thickness should be included.
Reply: Thanks for your reminding. The ear thickness was measured by electronic digital caliper (Deli, Zhejiang, China).
